# Feeling connected but dissimilar to one's future self reduces the intention-behavior gap

**Benjamin Ganschow**[1]*, **Sven Zebel**[2,3], **Jean-Louis van Gelder**[1,4], **Liza J. M. Cornet**[1]

**1** Institute of Education and Child Studies, University of Leiden, Leiden, Netherlands, **2** Department of Psychology of Conflict, Risk and Safety, University of Twente, Enschede, Netherlands, **3** Department of Private Law, Vrije Universiteit Amsterdam, Amsterdam, Netherlands, **4** Max Planck Institute for the Study of Crime, Security, and Law, Freiburg im Breisgau, Germany

* b.g.ganschow@fsw.leidenuniv.nl

**Data Availability Statement:** Data, procedures, analysis code, and supplementary analyses and materials are available online at: https://osf.io/h83nt/.

## Abstract

The intention-behavior gap is a common phenomenon where people fail to follow through on their intentions to change their behavior and pursue their future goals. Previous research has shown that people are more likely to act in favor of their future selves when they feel similar/connected to their future self and can vividly describe them. This study compared an imagination exercise with an integrated imagination and exposure exercise using virtual reality (VR) to embody age-morphed future selves to an imagination only exercise. We expected that strengthening the similarity/connectedness and the vividness of the future self would reduce the intention-behavior gap, and exposure to the future self would have the greatest effect. Surprisingly, the results showed that strengthening connectedness reduced the intention-behavior gap, but strengthening similarity increased the gap. Additionally, the exercises were equally effective in reducing the intention-behavior gap. These findings suggest that both feeling connected to and recognizing dissimilarity to one's future self play different roles in future-oriented behavior change.

## Introduction

The path towards an achievement tends to begin with a behavioral intention, e.g., "I am going to lose ten pounds" or "I want to tackle climate change!" [1]. However, intentions regularly fail to convert into action and slip into the 'intention-behavior gap' [2]. Meta-analyses indicate that roughly half of the people who intend to adopt health-related behaviors (e.g., attend cancer screenings, practice safe sex, increase physical activity) fail to follow through with their declared intentions [1, 3–5]. Additionally, the intentions themselves only account for around 30% of the variance in subsequent behavior change [1, 6, 7]. We propose and experimentally test a novel approach to reduce the intention-behavior gap. Our approach aims to reduce the gap by increasing the overlap in identity between the person expressing the commitment–the present self–and the person benefitting from the enacted intention–the future self. To accomplish this, we integrated two previously used methods that either had participants imagine their future self or exposed them to their age-morphed self-images in VR [8].

**Funding:** Initials: JLvG Grant number: 772911–CRIMETIME Full name: Jean-Louis van Gelder URL: https://erc.europa.eu/apply-grant/consolidator-grant No Text in manuscript: This study was funded by a European Research Council Consolidator Grant awarded to Jean-Louis van Gelder (Grant number 772911– CRIMETIME). The funders had no role in study design, data collection and analysis, decision to publish, or preparation of the manuscript.

**Competing interests:** The authors have declared that no competing interests exist. Powered

## Future self-continuity

People often intend to change their behavior in order to improve their well-being: to quit smoking, exercise, wake-up, or schedule regular visits to the dentist. These changes to behavior typically involve intertemporal choices that have immediate upfront cost and delayed reward (e.g., working out, abstaining from the pleasure of smoking). Multiple selves and self-continuity theorists argue that people are biased towards immediate rewards because people perceive their future self as different from their current self and thus are less motivated to imagine delayed consequences and act for the future self's well-being [9]. In support of this assumption, recent empirical research has demonstrated that a higher degree of overlap in identity (e.g., moral values, beliefs, behaviors, body type) between the present self and the future self, or future self-continuity, promotes future-oriented behaviors [9, 10].

Hershfield proposed three domains of how people use continuity with their future self to make choices: connectedness/similarity, vividness, and valence [11]. Firstly, people are motivated to work for people or organizations they share a relationship with, such as mother may give up a full-time job to care of her children. This strength of this motivation is a sum of the shared identities in the relationship (e.g., values, beliefs, personality, a shared history, purpose, preferences, etc.) [11]. Therefore, an index of the strength of a relationship between the present self and future self may be thought of as a sense of connection or of similarity [9, 12, 13]. Importantly, previous authors using this construct choose either future self-connectedness or similarity, as separate variables, or an average of both variables. However, they are expected to measure the same construct [14]. Secondly, people imagine outcomes in the future less vividly than outcomes closer to the present and tend underweight or ignore the future consequences if left unimagined [15]. Thus, people who fail to imagine their future self may underweight or ignore the consequences for their future self [8]. Lastly, people feel more connected to relationships they feel positive about. Thus, people who have a positive view of their future self may reinforce how connected they feel. Extant research finds that liking one's future self is related to current self-esteem, mental health, and general well-being [12, 14, 16]. However, an empirical connection between future-self valence and future-oriented behaviors is yet to be established and may be counterproductive—positive fantasies can give rise to the illusion that success will come easily and behavior change is not necessary [17]. Therefore, we expect that an increase in connectedness/similarity and vividness will help to reduce the intention-behavior gap, while valence is anticipated to have a null or negative effect.

## Strengthening future self-continuity through imagination and exposure

Exercises manipulating future self-continuity domains have participants either imagine their future self or expose them to a visual representation of their future self [8]. Imagination exercises require participants to explicitly think about their future self through vignettes [18] or short writing assignments [16, 19–21], or to elaborate upon and question their future self through guided imagination [16, 22]. Such exercises have been shown to reduce cheating behavior [23], increase savings behavior [18], reduce self-reported delinquency [21], increase physical activity [24], and reduce procrastination [22, 25]. However, they depend upon imaginative ability and motivation, which may vary considerably across respondents [26].

A set of alternative approaches to strengthening future self-continuity relies on visually exposing participants to age-morphed representations of their future self [8]. Previous research finds that participants exposed to their age-morphed future self also tend to make more future-oriented decisions. For instance, participants exposed to images of their age-morphed future self allotted more money to a retirement account [26, 27] and reported less delinquency [28]. These results also extend into VR interventions, where participants who embodied their

age-morphed avatar also saved more for retirement [26], cheated less [21], and engaged in less self-defeating behavior [29].

We argue that these two approaches may work in tandem because exposure bypasses motivation and ability dependent to imagination, while imagination has participants contemplate the specific identity of their future self. Additionally, embodiment of a future self avatar in virtual reality may aid the participants in imagining their future self and take their perspective. For instance, Slater et al. [30] created a self-counseling exercise to help people deal with personal issues. They had participants either switch between embodying a VR avatar representing themselves and an avatar representing Sigmund Freud, or to remain in the self-avatar while responding to a scripted version of Freud. Participants who embodied Sigmund Freud were more likely to report that they were "doing, thinking or feeling differently" and that the session had helped them with their personal issue. Similarly, exercises that have participants embody their future self, in addition to supplanting imagination and motivational deficits, may also adopt the characteristics of their future self. These two factors, imagination and VR embodiment of their age-morphed future self should therefore help participants feel an increased sense of future self-connection/similarity, and -vividness after completing the perspective taking exercise.

## Present study

The present study seeks to answer three questions: (1) Does the integration of VR embodiment with an imagination exercise strengthen future self-continuity domains more compared to an imagination-only exercise? (2) Can strengthening future self-continuity domains reduce the intention-behavior gap? (3) Does the exercise function through increases in future self-continuity? In other words, are the future self-continuity domains the mechanism that reduce the intention behavior gap in the exercise?

To accomplish this, we developed a perspective taking exercise [adapted from 31] to both strengthen the future self-continuity domains and also set a behavioral intention (see Methods). The exercise integrates the imagination and VR embodiment methods (Imagine and Embodied-VR condition) and was compared to an exercise that utilized only imagination in an *in vivo* lab setting (Imagine-IV condition) and to an exercise that utilized imagination in a VR environment (Imagine-VR condition). The Imagine-VR condition serves to control for placebo effects due to the difference in environment between *in vivo* and VR environments.

We expected that at least 50% of participants would act on their behavioral intentions consistent with the intention-behavior gap of well-being related behaviors reported in previous meta-analyses [1, 3, 4]. Second, we expected participants in the Imagine and Embodied-VR condition to strengthen future self-continuity domains more than the Imagine-IV or Imagine-VR conditions. Third, we also expected that participants who reported higher future self-connectedness/similarity, and -vividness post-exercise would act on their chosen intention more often, while -valence was expected to have no or negative influence. Lastly, we expected participants in the Imagine and Embodied-VR condition to be more likely to act on their intention than the Imagine-IV or Imagine-VR conditions, and that the increased likelihood in the Imagine and Embodied-VR condition would be mediated by increased future self- connectedness/similarity, and vividness post-exercise.

## Method

### Sample

We recruited 99 male university students by word of mouth and the faculty's recruitment panel from December to February 2019. The study was conducted concurrent with

recruitment. The VR software enabled the creation of male avatars only and females were not recruited. This limitation is due to the software being developed for incarcerated delinquents, who are largely male. We also excluded participants suffering from mental disorders or epilepsy because thinking about the future may be harmful for people with anxiety or depression, while epileptics may have adverse reactions to the visual features in VR. Participants received a €10 voucher or course credit in exchange for full participation in the study. Twenty-six participants per condition was determined by an a priori power analysis conducted with G*Power 3.1 [32] to detect a change in future self-continuity of $d = 0.5$ at an alpha of .05 and power of 0.8. This effect size was based upon previous manipulations of future self-continuity that resulted in behavior change ($d = 0.36$ to $0.77$) [16, 23, 31, 33]. We excluded eight participants who did not complete the last questionnaire and one due to mental health discovered on the day of study, leaving 90 participants (Imagine-IV = 27, Imagine-VR = 30, and Embodied-VR = 33; $M_{\text{age}} = 22.22$, $Range = 18–40$). The study was approved by the Ethics Committee of the Faculty of Behavioral, Management and Social Sciences at the University of Twente. (#191255). The individual in Fig 1 of this manuscript has given written informed consent to publish their likeness. This study's design and its analysis were not pre-registered. Data, procedures, analysis code, and supplementary analyses and materials are available online at: https://osf.io/h83nt/

## Materials

**Future self-continuity domains.** Future self-connectedness and -similarity were each measured using the overlapping circles scale [12]. The scale consists of a series of seven pairs of increasingly overlapping circles. The overlap between circle pairs signifies the degree of felt connection or similarity shared between present and future self in 10 years. Participants were asked to "Choose the pair of circles that you feel best represents the level of [similarity or

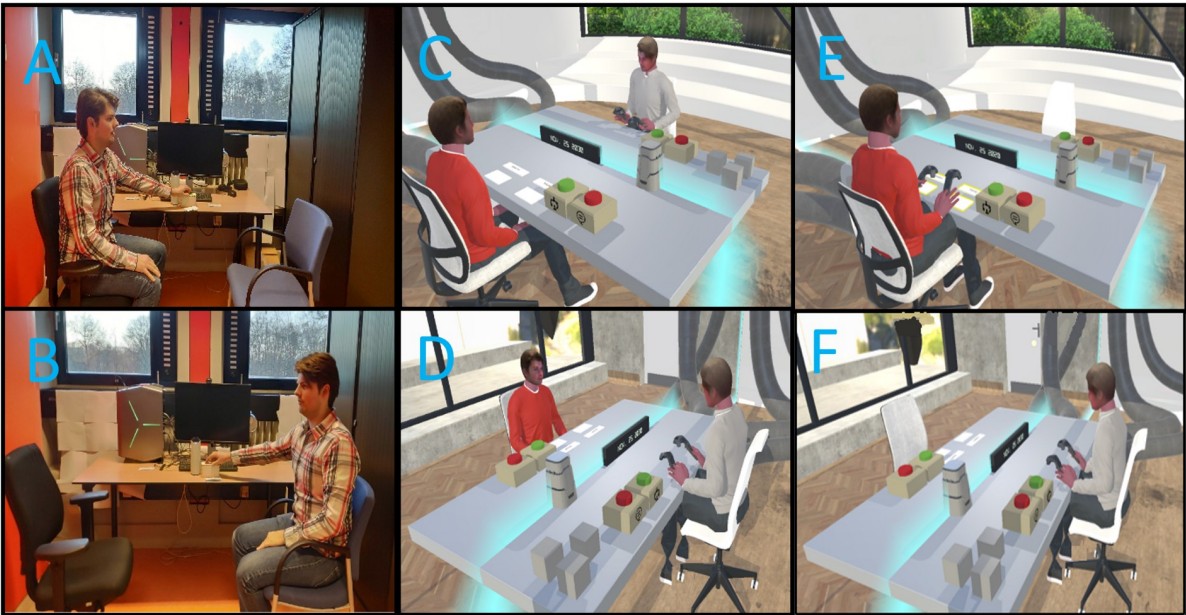

**Fig 1. Bird's eye view of the perspective taking exercise in the Imagine-IV, Imagine and Embodied-VR and Imagine-VR conditions.**
*Note.* Participants in Panels A and B are in the Imagine-IV, C and D in the Imagine and Embodied-VR and E and F in the Imagine-VR condition. Reprinted with permission from Benjamin Ganschow under a CC BY license, original copyright 2023.

connection] between your present self and yourself 10 years in the future." Reliability of their composite at baseline was low (α = .56) and the connectedness and similarity items were therefore analyzed apart. Future self-valence was measured by asking participants to rate how they "felt about their future self 10 years from now" [12] on a 7-point scale ranging from *don't like at all* to *like a lot*. Future self-vividness was measured with an abbreviated 3-item version of the 6-item Vividness of the Future Self Scale [28], i.e., "I find it easy to imagine myself 10 years from now", "I find it easy to describe myself 10 years from now", and "I do not have a clear image of myself 10 years from now" (reverse coded) on a 7-point scale ranging from *strongly disagree* to *strongly agree*. Reliability was high (α = .80).

**Vividness of Visual Imagery Scale (VVIQ).**   Vividness of Visual Imagery Scale (VVIQ) [34] was used to control for individual differences in imaginative ability relevant to performing the perspective taking exercise. Secondly, used in a post-hoc exploratory analysis to see if VR-embodiment aids low-imaginative ability [suggested in 26]. Participants were asked to imagine 16 scenes (e.g., "Visualize a rising sun.") and rate them on a 5-point scale from *No image at all* to *Perfectly clear and vivid as real seeing*. Reliability was high (α = .90) and the sum of the 16 items was used for analysis.

**Perspective taking exercise.**   The perspective taking exercise consisted of three phases: 1) choosing a successful future self, 2) 'two-chair' role-play between present and future self, and 3) setting a behavioral intention.

In the first phase, participants choose a long-term goal that their future self has achieved in 10-years. Participants explore possible long-term goals by writing for 5 minutes about one or more great accomplishments they would like to complete in 10 years' time and subsequently chose their favorite accomplishment (see S1 Appendix).

In the next phase, participants strengthen their connection to this future self through role-playing as their future self who accomplished the goal chosen in the first phase. The 'two-chair' role-play is a technique commonly used in clinical settings to help clients explore uncomfortable issues they are reluctant or ambivalent to discuss (e.g., addictions, traumas, binge-eating) by role-playing as a relation connected with the issue (e.g., affected children, their abuser) [35, 36]. Typically, the client sits across from an empty chair and role-plays a dialogue between the client and a significant other, who the client imagines is sitting in the empty chair. The client asks a question to the other, switches chairs, and answers from the perspective of this other. Clients are encouraged to get in character by switching between chairs, adapting mannerisms, and vocalizing questions. The two-chair was also adapted into VR as a self-counseling, with participants switching roles between client and the wise counselor Sigmund Freud [30, 37]. Taking perspective of their future self may improve participants motivation to imagine their long-term goals.

In our exercise, participants asked and answered 10 questions role-playing between their present self and their future self in 10 years who had completed the accomplishment they chose previously [31] (see S2 Appendix). Participants in the Imagine-IV condition began the exercise in the laboratory, seated in a chair across from an empty chair that they imagined their future self was sitting. They picked-up a question card and recorded the question to their future self (e.g., "How are you feeling today?"), and switched chairs. Now, as their future self, participants listened to the question, recorded an answer (e.g., "Great! I just finished my doctorate . . ."), then switched chairs and listened to the recording (see S3 Appendix).

In the Imagine-VR condition, the empty-chair exercise was re-created in a functionally equivalent VR environment: participants' controlled avatars in the 1st person, their avatars were sitting across from an empty chair, could pick up and read question cards, press buttons to record and play questions and answers, and press a button to switch chairs from the present to future and back.

In the Imagination and Embodied-VR condition, we added visual exposure and embodiment of the future self. We created avatars representing the present self by taking digital photos of the participant's face and used a computer software developed by Avatar SDK (www.avatarsdk.com) to render the facial features of the participant onto their avatar. For the future self avatar, an age-progression algorithm developed by Change My Face Ltd. (www.changemyface.com) was applied to the participant's avatar (see S3 Appendix). The avatar was age-morphed to look ten years older (see Fig 1). Before beginning the exercise, participants in the Imagine and Embodied-VR condition entered the VR environment as their present self and performed a brief embodiment exercise in front of a virtual mirror. Participants were instructed to look at the mirror and wave their hands and arms, move their torso, and shake their head [30].

In the final phase, participants chose a behavioral intention that would help them become the future self they previously role-played as. To help them choose, participants wrote down three future-benefiting behaviors and rated the time spent on the behavior, how many times a week this behavior would be performed, and how important the behavior was to becoming their future self. Participants then chose the behavior that they would intend to start in the next week (see S4 Appendix).

**Intention-behavior follow-up.** To follow-up on whether participants had acted on the behavioral intention they had chosen, participants were asked one week later if (1) they started their behavior (*yes/no)*, and if (2) they intended to continue the behavior (*yes/no)*. We also included a memory check to account for socially desirable answers in intention-behavior self-reports [6]. The check asked participants who answered they had continued their behavior to write down their behavioral intention and checked their answers to their previously chosen behavior. If participants could not correctly remember the behavioral intention they also intended to continue, they likely had not acted on their intention, and we classified them as not starting their behavior.

## Procedure

**Pre-experiment (T0).** At least one day before the experiment, participants completed an online questionnaire containing the future self-continuity and VVIQ scales, and the goal writing exercise. Participants then chose a time for the experimental session and were block-randomly assigned to one of the three conditions.

**Experiment (T1).** Participants were informed that they would be role-playing a conversation with their future self in 10 years who had accomplished the goal(s) that they had previously written about. The participants chose their favorite goal, and the researcher verified the goal would take around 10 years to achieve. The participants practiced the two-chair exercise until they could perform without help after which the researcher left. Once alone, the participants completed the 10 questions. Participants then filled out a post-test questionnaire with the future self-continuity scales, and the behavioral intention selection questions. The researcher asked questions about their experience, if they encountered any problems, and verified that the participants had chosen a behavioral intention that they could complete in the next week.

**Follow-up (T2).** One-week post-experiment, participants received an email with a link to an online questionnaire with the future self-continuity scales and the intention-behavior follow-up. If participants did not respond within one week, a second email was sent.

## Results

Descriptive statistics were computed in R [38] using the RStudio graphical interface [39]. Mediation analyses were computed with the PROCESS Macro in SPSS [40].

### The intention-behavior rate

At T2, out of 90 participants, 68 (75%) had begun their behavioral intention in the following week. Additionally, 66 (73%) of the 68 who started their behavior also indicated intending to continue afterwards. We used the more conservative 'starting and continuing' behavior as the intention-behavior rate. The memory check found that 15 of the 66 participants who intended to continue their behavior remembered a different behavior at T2 than they had chosen at T1. However, on further inspection, we found that the behavior recalled at the memory check matched one of the three alternate behaviors at T1 (see S1 File). The frequency of failing the memory check did not differ by condition. As such, participants may have changed their behavioral intention to an alternate behavior they later deemed better than their first choice. We conducted a check for robustness with these 15 participants excluded, which did not change the results (see S2 File). The analysis reported here therefore included all participants.

### Analytical strategy

First, we expect that 50% of participants would report acting on their behavioral intentions in the subsequent week. Second, predicted that the increase in the future self-continuity domains would be larger in the Imagine and Embodied-VR perspective taking exercise compared to the Imagine-IV or Imagine-VR exercises. Third, we predicted that within-person increases in future self-connectedness, -similarity, and -vividness would improve the intention-behavior rate, and future self-valence to weaken or have a null effect. Fourth, that the improvement in the intention-behavior rate from the Imagine and Embodied-VR condition would be mediated via future self-connectedness, -similarity and -vividness and expect positive indirect effects from these future self-continuity domains on the intention-behavior rate in the Imagine and Embodied-VR condition.

To test these hypotheses, we ran two models. The first is a logistic regression of condition on the intention-behavior rate. The second is was an ANCOVA multiple mediation analysis that calculates the direct effect of condition on intention-behavior rate, the direct effect of the within-person change in future self-continuity domains on the intention behavior-rate, and thirdly the indirect effect of the within person change that explains the variation from the condition on the intention-behavior rate. [41]. We specified into Model 4 of the PROCESS macro the intention-behavior rate as the outcome variable, the condition as the predictor, the future self-continuity domains at T1 as mediators, and T0 future self-continuity domains and VVIQ as covariates. These covariates predicted both mediator and outcome. Indirect effects were estimated from 5000 bootstrap estimates.

### Condition on intention-behavior rate

The intention-behavior rate was 78% for the Imagine-IV condition, 76% for the Embodied-VR condition, and 68% for the Imagine-VR condition. The logistic regression found no evidence for the expectation that participants in the Embodied-VR condition would have higher intention-behavior rates than those in the Imagine-IV and Imagine-VR conditions (see conditional model in Table 1). Post-hoc exploratory analysis found that these results also held while controlling for the conditional moderation of VVIQ, indicating that individual differences in imaginative ability did not influence the effectiveness of VR vs. non-VR conditions (see online S3 Appendix).

**Table 1. Future self-continuity domains predicting intention-behavior rate by condition.**

| | Conditional Model | | | Mediation Model | | | |
|---|---|---|---|---|---|---|---|
| | *exp(B)* | *p* | 95% CI | *exp(B)* | *p* | 95% CI | *VIF* |
| Imagine-IV | (ref) | | | (ref) | | | |
| Imagine-VR | 0.571 | 0.35 | 0.18, 1.87 | .027 | .101 | 0.06, 1.30 | 1.99 |
| I&E-VR | 0.893 | 0.85 | 0.27, 2.86 | 1.55 | .558 | 0.36, 6.71 | 1.99 |
| Connected T1 | | | | 2.21 | .005 | 1.26, 3.87 | 2.82 |
| Similarity T1 | | | | 0.46 | .010 | 0.83, -2.56 | 2.55 |
| Valence T1 | | | | 0.78 | .554 | 0.34, 1.78 | 1.27 |
| Vividness T1 | | | | 1.63 | .119 | 0.88, 3.01 | 1.95 |
| *Controls* | | | | | | | |
| Connected T0 | | | | 0.72 | .121 | 0.48, 1.09 | 1.91 |
| Similarity T0 | | | | 1.16 | .549 | 0.71, 1.91 | 1.74 |
| Valence T0 | | | | 0.96 | .881 | 0.55, 1.66 | 1.14 |
| Vividness T0 | | | | 1.19 | .880 | 0.69, 2.06 | 1.94 |
| VVIQ | | | | 1.28 | .549 | 0.57, 2.89 | 1.10 |
| *Model Diagnostics* | | | | | | | |
| AIC | 109.35 | | | 109.57 | | | |
| BIC | 116.85 | | | 139.57 | | | |
| Tjur's $R^2$ | .012 | | | .221 | | | |

*Note*: All coefficients are odds ratios. Imagine-IV condition was the reference category. VIF is variance inflation factor. AIC is Akaike Information Criterion. BIC is Bayesian Information Criterion. Imagine and Embodied VR is shortened to I&E VR.

## Condition on future self-continuity domains

From T0 to T1, all conditions tended to increase participants' vividness and valence. Additionally, the Imagine-VR condition also increased connectedness, whereas the Imagine and Embodied-VR condition also increased similarity (see Table 2).

Results from the mediation model found that the within-person change in future self-continuity domains at T1 did not vary by condition.

## Future self-domains on intention-behavior rate controlling for condition

Results show a positive effect on the intention-behavior rate from connectedness at T1 ($b = 0.79$, $Z = 2.78$, $p = .006$, $OR = 2.2$, 95% CI [1.26, 3.87]) that partly confirms the future self-continuity hypothesis. However, the negative effect of similarity at T1 ($b = -0.77$, $Z = -2.56$, $p = .010$, $OR = 0.46$, 95% CI [0.26, 0.83]) runs opposite to the hypothesized direction (see Table 1 and Fig 2). Lastly, unexpectedly, there was a positive but non-significant effect of vividness at T1 ($b = 0.49$, $Z = 1.56$, p = .119, $OR = 1.63$, 95% CI [0.88, 3.01]).

## Mediation of conditional effect through future self-domains

Previous analysis neither a conditional direct effect on the outcome (intention-behavior rate) nor on the mediators (future self-continuity domains), and therefore also no indirect effect (see online mediation results). These analyses indicate no support for our first two predictions that participants in the Imagine and Embodied-VR condition are more likely to act on their intention than the Imagine-IV or Imagine-VR conditions or that the likelihood would be mediated by change in future self- connectedness, -similarity, or -vividness.

**Table 2. Means, standard deviations, and effect sizes of the future self-continuity domains at and from T0 and T1.**

| | Condition | N | T0 | | T1 | | Effect Size | |
|---|---|---|---|---|---|---|---|---|
| | | | **M** | **SD** | **M** | **SD** | **d** | **95% CI** |
| Connected | Imagine-IV | 27 | 4.6 | 1.7 | 5.0 | 1.5 | 0.25 | -0.12, 0.61 |
| | Imagine-VR | 30 | 4.0 | 1.6 | 5.0** | 1.5 | 0.58 | 0.24, 0.99 |
| | I&E-VR | 34 | 4.4 | 1.9 | 4.4 | 1.6 | 0.02 | -0.37, 0.36 |
| Similarity | Imagine-IV | | 3.8 | 1.6 | 4.1 | 1.6 | 0.22 | -0.15, 0.60 |
| | Imagine-VR | | 4.4 | 1.4 | 4.0 | 1.4 | -0.23 | -0.69, 0.12 |
| | I&E-VR | | 3.9 | 1.3 | 4.4 • | 1.5 | 0.34 | 0.00, 0.70 |
| Valence | Imagine-IV | | 6.0 | 1.0 | 6.4* | 0.9 | 0.42 | 0.03, 0.79 |
| | I&E-VR | | 5.6 | 1.0 | 6.2** | 0.6 | 0.50 | 0.15, 0.92 |
| | Embodied-VR | | 5.7 | 1.2 | 6.2* | 0.9 | 0.40 | 0.04, 0.89 |
| Vividness | Imagine-IV | | 3.5 | 1.6 | 4.3*** | 1.3 | 0.81 | 0.44, 1.44 |
| | Imagine-VR | | 3.6 | 1.3 | 4.2** | 1.2 | 0.60 | 0.26, 1.05 |
| | I&E-VR | | 3.4 | 1.5 | 4.0** | 1.12 | 0.45 | 0.11, 0.87 |

*Note*: Asterisks represent significance levels for pairwise t-tests between T0 and T1 where • $p < .10$, * $p < .05$, ** $p < .01$, *** $p < .001$. Imagine and Embodied VR is shortened to I&E VR.

## Discussion

Intentions are the first step towards behavior change, but many intentions are left untried [2]. In this study, we aimed to increase the intention-behavior gap by strengthening participants' future self-continuity domains. 90 male participants role-played a conversation as their successful future self in 10 years time. The exercise was completed either through imagination, through imagination plus virtual embodiment of their age-morphed future self, or through imagination in a placebo-VR environment without embodiment. After the exercise, participants chose a behavioral intention they would begin in the next week that would benefit their future self. A week later, participants were asked if they had acted on and intended to continue the behavior, which constitutes the intention-behavior rate. We expected positive changes in future self-continuity domains of connectedness/similarity and vividness to improve the intention-behavior rate. We also expected that combining an imaginal and VR embodiment exercise would reduce the intention-behavior rate the most. Finally, we expected the effect on the intention-behavior rate from the integrated embodiment exercise to be mediated by the future self-continuity domains.

The first important finding is that within-person increases in connectedness improved the intention behavior rate while, contrary to expectations, within-person increases in future self-similarity reduced it. Vividness appeared as a positive, but non-significant predictor. The second important finding is that on average, 73% of participants who completed any perspective taking exercise reported that they followed through with their intentions. Thirdly, and unexpectedly, visual embodiment through VR provided no added benefit to the perspective taking exercise to either increase the future self-continuity domains or improve the intention-behavior rate.

### Strengthening the future self-continuity domains on intention-behavior rate

Many studies combine future self-connectedness and -similarity as a composite as we attempted to, but the two measures made for an unreliable composite and we therefore chose

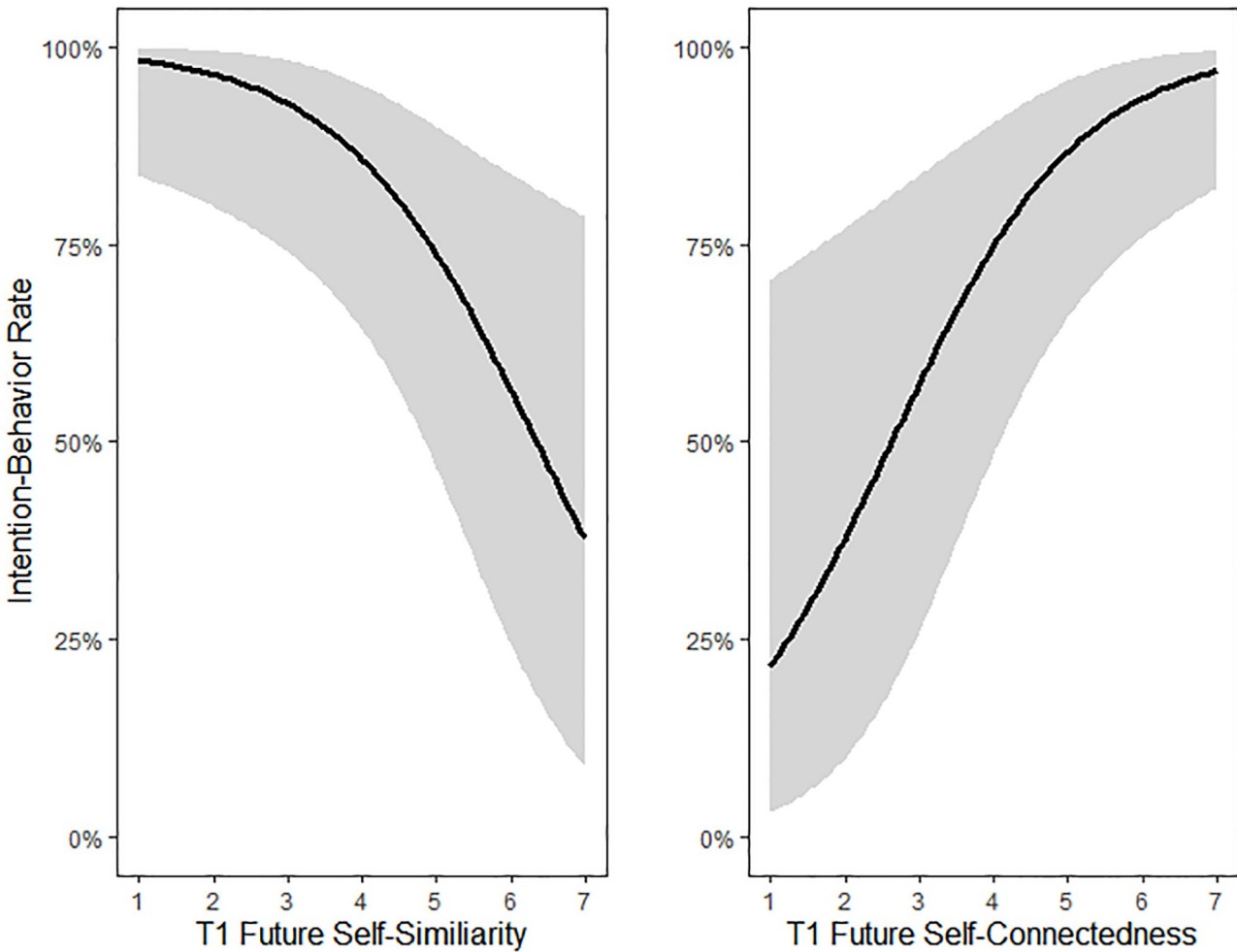

**Fig 2. Relation of T1 future self-similarity and -connectedness to the intention-behavior rate.** *Note*. Intention-behavior rates were predicted while controlling for T0 future self-continuity domains and VVIQ. Shaded areas represent 95% confidence intervals.

to treat them separately. In doing so, we found that when starting new behaviors, participants were more likely to act when they felt more connected to their future selves, which confirms its role in motivating future-oriented behavior change. Unexpectedly however, participants acted less on their intentions when they felt more similar to their future self. This surprising result adds new insight to the conflicting empirical evidence that future-oriented behaviors are more likely when people recognize both present to future 'self-gaps' and 'self-continuity' [42].

In the future self-gaps accounts, such as discrepancy reduction [43] or possible selves [44], people perform future-oriented behaviors to reduce the perceived gap between their present self and an imagined positive future. Goal-setting exercises often emphasize self-gaps through mental contrasting present and future selves to identify relevant gaps [17]. In contrast, the future self-continuity account predicts that people perform future-oriented behaviors when they perceive more overlap in identity, or less of a gap, between present and futures selves. For instance, Peetz and Wilson [45] found that participants who felt dissimilar from a healthy future self motivated them towards more intentions to exercise and choose healthier food rewards. Although the perspective taking exercise was designed to strengthen both future self-similarity and–connection, it also likely induced dissimilarity because we had participants

think of the difficulties in becoming their future self. Our results may support both accounts, suggesting that participants were motivated to act on their intentions by recognizing the gaps, or dissimilarity, between present and future selves, as well as by feeling more connected.

These surprising results points towards interesting avenues for future studies. Firstly, this study was not designed to investigate the self-continuity and self-gap accounts using future self-connectedness and -similarity and our results should be replicated with scales designed to better disambiguate self-continuity from self-gaps. Secondly, our evidence suggests that participants' self-continuity and self-gaps may be used in tandem to change behaviors and future research could investigate how people use either self-gap and self-continuity strategies to influence future-oriented behavior.

## Imagination, embodiment, and the intention-behavior rate

Around three-fourths of participants acted on their new intentions in the week following the perspective taking exercise. This overall intention-behavior rate compares favorably to the roughly equal chance of intentions turning into behavior reported in previous meta-analyses on health and exercise behaviors [1, 3, 4]. Additionally, of all the participants who chose to start their behavior, only two did not wish to continue their intentions, which suggests stability of intentions, which is among the strongest moderators predicting continued behavior [46].

The imaginal perspective-taking exercise with added embodiment of age-morphed VR avatars did not improve intention-behavior rate more or lead to noticeable improvements to the future self-continuity domains in comparison to the non-embodied conditions. These findings show that enacting behavior change may not need a visual or embodiment aid to improve the likelihood of participants following through on new behaviors. Previous results did observe greater change in savings after exposure or to age-morphed avatars or more intention to exercise after embodying weight-reduced avatars compared to non-age morphed conditions [21, 26, 47]. This could be because the behaviors studied are more strongly related to the avatar. It may also be because the behavior was measured immediately after the exposure when the future self was salient. In this study, intentions may not be linked to age-morphing and the choice to act on the intention occurred in the following week and the exposure to age-morphing may not have remained salient during this decision. Future interventions using exposure or age-morphing may seek to elicit a stronger relationship between exposure and behavior or benefit from either repeated exposure, for example by using a smartphone application on a daily basis [see e.g., 48], or to train participants to recall the age-morphing of their future self during decisions.

The main limitation to this study is our study may be underpowered detect the effect size that our exercises would need to manipulate the intention-behavior rate. Therefore, we used previous studies that manipulated future self-similarity or -connectedness to approximate the effect size for our sample size calculations. This is particularly the case for future self-valence because participants tended to like their future selves and there was little variance between T0 and T1. This result is also seen in other studies [12, 14, 31] and future research should determine sample size with a smaller effect sizes. Secondly, the follow-up was conducted only a week after the experiment, which is enough to investigate the transition from intention to behavior, but not enough time to draw conclusions about long-term adoption of the behavior, or if VR embodiment or non-embodied exercises differentially influenced adoption. Lastly, we had two practical constraints that the limited interpretation of our conclusions. The first we could only include male participants. We chose to use a non-embodied VR condition rather than an inert control condition because previous studies using VR did not control for the placebo effect of being in VR and test subjects used during the development of the exercise

favored the Imagine and Embodied-VR condition over the *in vivo* condition This practical decision limited our conclusions and we compared the intention-behavior rate to previous studies.

This limitation is also a strength of this study, because the VR embodiment literature lacks studies that measure the placebo effect from the VR environment, or additionally, comparing an *in vivo* exercise to the implementation in VR [49]. These comparisons are necessary to better understand which participants may benefit from embodied VR implementations (e.g., those with low imaginative ability or motivation). In this case, our design allowed us to conclude that the novelty of the VR environment or embodied VR neither hindered nor particularly benefited participants in comparison to the *in vivo* exercise. Lastly, the perspective taking exercise is relatively simple to implement and can be completed in stand-alone and remote fashion.

## Conclusion

This study demonstrates that with only about half an hour, two chairs, and a future self, male participants were more than likely to follow-through on their intentions–where we expected only about half of participants to do this. Participants were, however, equally likely to act on their intention if they performed the exercise using their own imagination or whether their imagination was augmented by embodying a VR avatar of their future self. Our participants were more likely to start their behavior when they felt more connected and dissimilar to their future self. Bridging the intention-behavior gap is possible when recognizing the continuity that binds the future to the present, but also the gap the divides the present from the future, which combine in a call for action.

## Supporting information

**S1 Fig. Bird's eye view of the perspective taking exercise in the Imagine-IV, Imagine and Embodied-VR and Imagine-VR conditions.** *Note.* Participants in Panels A and B are in the Imagine-IV, C and D in the Imagine and Embodied-VR and E and F in the Imagine-VR condition. Reprinted with permission from Benjamin Ganschow under a CC BY license, original copyright 2023.
(TIF)

**S2 Fig. Relation of T1 future self-similarity and -connectedness to the intention-behavior rate.** *Note.* Intention-behavior rates were predicted while controlling for T0 future self-continuity domains and VVIQ. Shaded areas represent 95% confidence intervals.
(TIF)

**S1 Appendix. Writing assignment.**
(DOCX)

**S2 Appendix. Perspective taking exercise questions.**
(DOCX)

**S3 Appendix. Technical details.**
(DOCX)

**S4 Appendix. Behavior selection questions.**
(DOCX)

**S1 File. Behaviors chosen by participants.**
(XLSX)

**S2 File. Robustness check.**
(DOCX)

## Author Contributions

**Conceptualization:** Benjamin Ganschow, Sven Zebel, Jean-Louis van Gelder, Liza J. M. Cornet.

**Data curation:** Benjamin Ganschow.

**Formal analysis:** Benjamin Ganschow, Sven Zebel.

**Funding acquisition:** Jean-Louis van Gelder.

**Investigation:** Benjamin Ganschow, Sven Zebel, Jean-Louis van Gelder, Liza J. M. Cornet.

**Methodology:** Benjamin Ganschow, Sven Zebel, Jean-Louis van Gelder, Liza J. M. Cornet.

**Project administration:** Benjamin Ganschow, Jean-Louis van Gelder, Liza J. M. Cornet.

**Resources:** Benjamin Ganschow, Jean-Louis van Gelder.

**Software:** Benjamin Ganschow, Jean-Louis van Gelder.

**Supervision:** Benjamin Ganschow, Sven Zebel, Jean-Louis van Gelder, Liza J. M. Cornet.

**Validation:** Benjamin Ganschow, Liza J. M. Cornet.

**Visualization:** Benjamin Ganschow.

**Writing – original draft:** Benjamin Ganschow, Sven Zebel, Jean-Louis van Gelder, Liza J. M. Cornet.

**Writing – review & editing:** Benjamin Ganschow, Sven Zebel, Jean-Louis van Gelder, Liza J. M. Cornet.

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
