## [Decision Letter · Decision Letter 0]

5 Nov 2023

PONE-D-23-19069Feeling connected but dissimilar to one’s future self reduces the intention-behavior gapPLOS ONE

Dear Dr. Ganschow,

Thank you for submitting your manuscript to PLOS ONE. After careful consideration, we feel that it has merit but does not fully meet PLOS ONE’s publication criteria as it currently stands. Therefore, we invite you to submit a revised version of the manuscript that addresses the points raised during the review process. Overall, your manuscript is well written and brings a valuable contribution to this area of research. Two reviewers made interesting comments and suggestions that should be considered during the preparation of your revised version. You can check both of the reviews at the end of this message.

We look forward to receiving your revised manuscript.

Kind regards,

André Rabelo, Ph. D.

Academic Editor

PLOS ONE

Journal Requirements:

2. We note that Figures 1C, 1D, 1E and 1F  in your submission contain copyrighted images. All PLOS content is published under the Creative Commons Attribution License (CC BY 4.0), which means that the manuscript, images, and Supporting Information files will be freely available online, and any third party is permitted to access, download, copy, distribute, and use these materials in any way, even commercially, with proper attribution. For more information, see our copyright guidelines: http://journals.plos.org/plosone/s/licenses-and-copyright.

a. You may seek permission from the original copyright holder of Figures 1C, 1D, 1E and 1F to publish the content specifically under the CC BY 4.0 license. 

4. We note that Figures 1A and 1B includes an image of a participant in the study. 

Reviewers' comments:

Reviewer's Responses to Questions

**Comments to the Author**

1. Is the manuscript technically sound, and do the data support the conclusions?

Reviewer #1: Yes

Reviewer #2: Partly

2. Has the statistical analysis been performed appropriately and rigorously? 

Reviewer #1: Yes

Reviewer #2: Yes

3. Have the authors made all data underlying the findings in their manuscript fully available?

Reviewer #1: Yes

Reviewer #2: Yes

4. Is the manuscript presented in an intelligible fashion and written in standard English?

Reviewer #1: Yes

Reviewer #2: Yes

5. Review Comments to the Author

Reviewer #1: The current study focused on the impact of various future-self perspective-taking modalities on changes in future self-continuity (FSC) domains, as well as subsequent intention to perform future-oriented behaviors. I enjoyed reading the manuscript and believe that the topic has both theoretical and practical importance. I particularly want to highlight the experimental design of the study (e.g., having the multiple VR conditions and the in-vivo control) and the longitudinal nature of the data collection (e.g., measuring and controlling for baseline FSC and following up with participants regarding their intentions-behaviors).

I did have some questions and comments as I went through the manuscript that I believe would help improve any revision. These items are listed below based on when I had them as I was going through the document. As a result, they are not ordered based on perceived importance.

1. Line 192. It says the average of the 16 VVIQ items were used, but based on the SPSS data file included on OSF it appears that sum/total scores were used (but I could be looking at the wrong column). If there is no missing data on the items (which appears to be the case), the results using mean/average or sum/total scale scores will be the same. But you would still want to be clear if average or total scale scores were used.

2. Lines 227-228. For the Figure 1 caption, I believe that the first “Imagine-VR” should be “Imagine-IV”.

3. Follow-up: Did everybody respond to the behavioral intention follow-up, or was there attrition? It doesn’t seem like there was attrition because on line 273 the full sample size of 90 is listed as the denominator. Or, were the 8 participants mentioned on lines 163-164 as having incomplete data people that did not respond to the follow-up? Was there any incentive for participants to complete the follow-up? In my experience it is very difficult getting such a high percentage of the sample to respond to even a quick follow-up, so any information detailing how the follow-up response rate was achieved would be helpful.

4. On lines 279-281 it is stated “The frequency of failing the memory check did not differ by condition (see Supplementary Materials 1).” I do not see a condition column/variable included in these supplementary materials, so the veracity of this statistical statement cannot be confirmed in the supplementary materials.

5. Lines 309-310. For people not familiar with the PROCESS macro, it would be helpful to make it clear that the T0 future self-continuity and VVIQ covariates are predicting both the mediators and the intention-behavior outcome in the model. In some specifications of mediation models, covariates only predict the outcome and not the mediator(s) (or vice versa). I have no issue with the specification of the model here per se, but just think this clarification would be helpful for a reader (so that they have a clear image of the mediation path model being tested).

6. Line 313. A bit nitpicky but you say proportion but then provide percents. Similarly, in Line 362 they are described as probabilities and not percents.

7. Line 317-320 reference a moderation analysis and then says to “see online supplementary analyses”. I was not able to locate where these analyses are on OSF, so a clearer description of the location would be needed.

8. I believe the labeling of the first row in Table 1 will confuse readers. I think “Imagine-IV” in the first row is actually the intercept/constant of the logistic regression models, correct? Because Imagine-IV is the reference group you usually don’t expect to see coefficients associated with that group. Now, because they are the reference group, it is the case that you can interpret the intercept as the intention-behavior rate for the Imagine-IV group (and the adjusted intention-behavior rate after controlling for the mediators and covariates in the mediation model). However, I believe this subtle detail will be lost on many readers, and I feel many might interpret the significant p-value in the conditional model as somehow implying that the Imagine-IV condition had higher odds of intention-behavior than the other groups.

9. Additionally, in Table 1, why is the p value for this first row .901 when the 95% CI for the odds ratio is completely above 1?

10. I was confused by lines 315-316. It says the expectation was that the Embodied-VR condition would reduce the intention-behavior “rate” more than the other groups. I know the hypothesis was that the Embodied-VR condition would reduce the intention-behavior gap, but for the rate doesn’t a higher rate imply less of a gap? That is, wouldn’t it be expected that the Embodied-VR condition would exhibit the highest rate? A similar issue is with the wording in Line 428.

11. Table 2. Not clear to me what the grey line towards the bottom of the table represents (separating the conditions for the vividness item). But this could just be a line added to the pdf document on my end.

12. Looking at the data in Table 2, it does look like there was a bit of a ceiling effect for the valence item. There still was enough room for significant increases to be observed from T0 to T1, but looking at the Ms and SDs it is still the case that a lot of participants were likely at the highest point of 7 at baseline. Because the T0 FSC covariates also predicted the T1 FSC mediators, the mediators essentially act as a measurement of the within-subject change in the FSC constructs. As a result, there could have been reduced power to observe an effect of valence due to the inability to observe potential increases in valence for the percentage of participants that were at ceiling at baseline. I am not necessarily predicting an effect of valence would have been found otherwise, but it would be helpful to at least reference the potential ceiling effect on the valence item in the Discussion.

13. In lines 337-338, it is mentioned that the PROCESS macro ran four logistic regressions. I think the wording here might confuse some readers. To my understanding based on the earlier description of the mediation model and the PROCESS image included on OSF, one mediation model was performed with all 4 mediators included. That does mean four separate set of coefficients will be provided from the condition IV to the mediators, but some readers might interpret the wording here to imply that the mediation model was performed 4 times (with one mediator included at a time). Second, why would these predictions of the mediators be described as logistic regressions since the FSC mediators presumably are being treated as continuous variables (which is confirmed in the PROCESS image on OSF since the mediators are scale variables)?

14. In Lines 448-449 the results of a pilot study are referenced for the first time. It was a little confusing as a reader because when I read that line it forced me to go back to the Methods/Results to see if that pilot study was mentioned before. I am not implying you need to reference those results before, but you may want to introduce the pilot study in these lines better so that it doesn’t confuse a reader.

15. In the Supplemental 2 Results, sometimes different condition names are used compared to the main document (e.g., Visual-VR). I recommend keeping the names consistent so it is clear to a reader comparing both results.

Reviewer #2: This study compared different ways of imagining a future self being successful with a specific focus on imagination and virtual reality integration. The authors looked at whether the impact of exposure to different future imaginings, and resultant ratings of similarity, connectedness and vividness to that future self on the transfer of intentions into behaviour.

The set up for the study was interesting and the authors offer sound possible interpretations of their findings. Where the study could be improved, from my perspective, is in the presentation of research questions and methodology; I found it difficult to follow this study and efforts could be put into place to more clearly explain what is expected and how the study was conducted. I provide more specific details below.

Introduction:

I found the introduction to be well written, logical and clear.

When describing the present study, I am curious why described the future self as having completed a long-terms personal goal (versus not having completed them). That is, why did the authors choose to take an approach versus an avoidance perspective. When imagining future selves, people may pursue a future self they would like to become (e.g., get my doctorate) but they may also want to engage in a behaviour (e.g., exercise) to avoid a future self they don’t want to realise (e.g., develop a chronic condition, get out of shape etc.). The chose to focus on a successful future self could be unpacked and justified.

Method

It seems strange and exclusive that the VR software only allowed for the creation of male avatars. It begs the question, why are there VR software out there that are made to represent only one gender? The choice of this VR software limits the generalizability of the study, which is a big limitation.

What inspired or served as the basis for the measurement of future self-continuity domains?

The authors explain why they controlled for imaginary ability by saying that “VR may aid participants with low imaginative ability”. This doesn’t see like a reason to control for imaginative ability. Wouldn’t you control for imaginative ability because it may influence how well one engages with imaginative tasks?

The authors should provide an explanation of what the steps of the perspective taking exercise are meant to accomplish. For example, for step 1, why did they have participants describe their future self having achieved a goal versus not having achieved it? For step 2, what do the authors think will come of the two chair exercise? And in step three, the behavioural intention? The reader can try to infer why these steps were chosen and what they are meant to accomplish but it would be clearer if the authors were explicit about why they chose these activities.

In the Imagine-VR condition, I understand that the participants did not use avatars, but then what did the virtual experience look like? Who did they see? What aspects about it were VR and why was this necessary?

Overall, what was the intended function of the two-chair exercise? This is implied but could be made more explicit.

Presumably, step 3 – setting an intention – was measured to test if the other two steps impacted the intention behaviour gap. Just be clear about this – help pull together the different pieces of the study so that it is easy for the reader to understand.

Were participants told that their goal would take about 10 years to achieve? Why was this an aspect of the study? Why didn’t this get any mention earlier when describing the intervention? Couldn’t participants have chosen something that could be achieved and maintained such as getting in shape, maintaining a healthy body weight, being in a happy marriage etc.? What if the goal was not verified as taking 10 years to achieve? What happened then?

Description of the conditions happens in bits and pieces throughout the methods which is confusing. I think it would be easier for the readers to understand if a clear description of the conditions were given all at once, in one place.

What did participants in the VR condition see? I don’t understand this brief embodiment exercise described involving a virtual mirror. Readers should not be expected to go read reference 31 to get an understanding of this condition.

The recording aspect of the two chair method is unclear to me. What does it mean that a participant recorded a question and then answer to a question. Did they have the recording played back to them once they were done? Couldn’t they just hear themselves ask and answer in real time? Why was a recording necessary?

Was anything done to ensure that the intervention conditions were carried out as intended?

Results

Early in the results, mediation analyses are mentioned for the first time. No mention of mediation was set up in the intro – why test for mediation and what is the proposed mediator? It appears from the hypotheses that moderation may be something tested (though if so, that should be explicitly stated in the hypotheses) insomuch as stronger self-continuity is supposed to be associated with a reduction in the intention-behaviour gap.

On page 19 (marked page 13) , there is mention of a hypothesis that “… the increased likelihood in the Imagine and Embodied-VR condition would be mediated via future self-connectedness/similarity and vividness. Yet this is not clearly outlined in the study research questions listed on the bottom of page 6. There should be clear continuity across the paper in terms of the research questions and analyses. This is lacking.

In general, it was not clear to me by the time I got to the results what analyses were going to be carried out and, relatedly, what types of relationships (mediations) were expected. The results should flow naturally and expectedly from the set up earlier in the introduction. That is lacking here.

The authors discuss studies where VR embodiments have led to change – and cite weight loss studies. It may be that the VR embodiment may have to pertain to some outcome (like weight loss) that can be visually represented for the embodiment aspect to be motivational. In this study, the age-morphed avatar would not embody any of the changes (especially because people choose their own changes, so it wasn’t uniform and incorporated). One might think that the use of avatars may even be detrimental as people age. I am in my mid 40’s and seeing a 10-year age-morphed avatar may be nothing but depressing… something to consider. Bottom line – if the age morphed avatar doesn’t depict the thing that the person is seeking to change (which it wouldn’t with something like, finished my doctorate) the virtual aspect may not offer anything over one’s imagination – unless the avatar was going through the motions of that future self (e.g., walking through the halls of a university, conducting research etc.).

I was surprised to not see the rather large fact that only one gender was represented (without any rationale for the limitation) not listed as a limitation. I understand that this was a logistical necessity based on the VR program – but the choice to use that program is a limitation that should be acknowledged. Further the findings are not discussed with any acknowledgment to the fact that they pertain only to a sample of males. This major limitation should be acknowledged and discussed. One example of how this limitation could be acknowledged is if instead of saying “participants” in the discussion, the word “male participants” is used.

Minor points:

Page 13 (of PDF), line 154 under Method – something is off in the wording of this sentence.

Page 15 (of PDF), line 213. I think the word “sat” should be removed.

Page 25 (of PDF), discussion, line 412: result should be results or it should read “This suprising result”.

6. PLOS authors have the option to publish the peer review history of their article (what does this mean?). If published, this will include your full peer review and any attached files.

Reviewer #1: No

Reviewer #2: No

---

## [Author Response · Author response to Decision Letter 0]

26 Mar 2024

Dear editor and reviewers,

 Thank you for your attention to detail and inciteful comments while reviewing our work. We agree with all of the points raised and have integrated most commentary into our revised manuscript. We comment after each point with an answer to the issue, how we attempted to address the it, and where in the manuscript.

 Found that both reviewers had trouble following how the results followed from the hypothesis and the major changes were to the presentation of the hypothesis, the method, the results, and to the online materials.

We hope these changes are satisfactory and we look forward to your commentary.

Reviewer #1

The current study focused on the impact of various future-self perspective-taking modalities on changes in future self-continuity (FSC) domains, as well as subsequent intention to perform future-oriented behaviors. I enjoyed reading the manuscript and believe that the topic has both theoretical and practical importance. I particularly want to highlight the experimental design of the study (e.g., having the multiple VR conditions and the in-vivo control) and the longitudinal nature of the data collection (e.g., measuring and controlling for baseline FSC and following up with participants regarding their intentions-behaviors).

I did have some questions and comments as I went through the manuscript that I believe would help improve any revision. These items are listed below based on when I had them as I was going through the document. As a result, they are not ordered based on perceived importance.

1. Line 192. It says the average of the 16 VVIQ items were used, but based on the SPSS data file included on OSF it appears that sum/total scores were used (but I could be looking at the wrong column). If there is no missing data on the items (which appears to be the case), the results using mean/average or sum/total scale scores will be the same. But you would still want to be clear if average or total scale scores were used.

We originally used sum scores in the last version of the paper but we thought that the VVIQ coefficient from a mean composite is easier to interpret. The results from table 2 are from line 202 in the R file: “effectsizes+scalereliabilities+descriptives+spsscheck_fscintentionbehavior.R”. We have updated the spss syntax to correct this confusion.

Changed at line 175

2. Lines 227-228. For the Figure 1 caption, I believe that the first “Imagine-VR” should be “Imagine-IV”.

This was changed in line 225. 

3. Follow-up: Did everybody respond to the behavioral intention follow-up, or was there attrition? It doesn’t seem like there was attrition because on line 273 the full sample size of 90 is listed as the denominator. Or, were the 8 participants mentioned on lines 163-164 as having incomplete data people that did not respond to the follow-up? Was there any incentive for participants to complete the follow-up? In my experience it is very difficult getting such a high percentage of the sample to respond to even a quick follow-up, so any information detailing how the follow-up response rate was achieved would be helpful.

The low attrition was likely due 10$/course credit upon filling out a short-5 minute online questionnaire a week later and we also followed-up a week later with an additional email to the participant (around 90% of participants responded to first email). We have exact numbers in our data of how many participants needed a follow-up. The 8 respondents did not fill out that questionnaire. We changed the language the “We excluded eight participants who did not complete the last questionnaire” to be clearer in where the attrition occurred. 

4. On lines 279-281 it is stated “The frequency of failing the memory check did not differ by condition (see Supplementary Materials 1).” I do not see a condition column/variable included in these supplementary materials, so the veracity of this statistical statement cannot be confirmed in the supplementary materials.

We changed where supplementary materials 1 was referenced to not include the differences between groups in attrition to not include the difference in memory check. We added a table and the chi-square test at the beginning of supplementary materials 2.

5. Lines 309-310. For people not familiar with the PROCESS macro, it would be helpful to make it clear that the T0 future self-continuity and VVIQ covariates are predicting both the mediators and the intention-behavior outcome in the model. In some specifications of mediation models, covariates only predict the outcome and not the mediator(s) (or vice versa). I have no issue with the specification of the model here per se, but just think this clarification would be helpful for a reader (so that they have a clear image of the mediation path model being tested).

Good point. We wanted to fully control for the baseline (T0) values for both the mediator and outcome. We added a statement clarifying this at line 301.

6. Line 313. A bit nitpicky but you say proportion but then provide percents. Similarly, in Line 362 they are described as probabilities and not percents.

Thanks for noticing. We changed the deleted the proportion at line 313 because “intention-behavior rate” is a statistic and probabilities at in 362 to “intention-behavior rates”

7. Line 317-320 reference a moderation analysis and then says to “see online supplementary analyses”. I was not able to locate where these analyses are on OSF, so a clearer description of the location would be needed.

This is in the r-code and really a quagmire to find. We include a new appendix.

8. I believe the labeling of the first row in Table 1 will confuse readers. I think “Imagine-IV” in the first row is actually the intercept/constant of the logistic regression models, correct? Because Imagine-IV is the reference group you usually don’t expect to see coefficients associated with that group. Now, because they are the reference group, it is the case that you can interpret the intercept as the intention-behavior rate for the Imagine-IV group (and the adjusted intention-behavior rate after controlling for the mediators and covariates in the mediation model). However, I believe this subtle detail will be lost on many readers, and I feel many might interpret the significant p-value in the conditional model as somehow implying that the Imagine-IV condition had higher odds of intention-behavior than the other groups.

We agree that it is confusing. Imagine-IV was the reference category. We removed the intercepts and placed (ref) to avoid distractions. 

9. Additionally, in Table 1, why is the p value for this first row .901 when the 95% CI for the odds ratio is completely above 1?

The confidence intervals are wrong. The We reran the analyses. confidence intervals of the intercept were inflated by the program used to convert the log-odds to odds ratios and calculate confidence intervals. We double-checked a few other coefficients in the model but they are correct.

10. I was confused by lines 315-316. It says the expectation was that the Embodied-VR condition would reduce the intention-behavior “rate” more than the other groups. I know the hypothesis was that the Embodied-VR condition would reduce the intention-behavior gap, but for the rate doesn’t a higher rate imply less of a gap? That is, wouldn’t it be expected that the Embodied-VR condition would exhibit the highest rate? A similar issue is with the wording in Line 428.

Yes, it should increase the rate and hence a lower gap. We amended this at 428 

11. Table 2. Not clear to me what the grey line towards the bottom of the table represents (separating the conditions for the vividness item). But this could just be a line added to the pdf document on my end.

That is a formatting issue with the .pdf. It is also present on our .pdf.

12. Looking at the data in Table 2, it does look like there was a bit of a ceiling effect for the valence item. There still was enough room for significant increases to be observed from T0 to T1, but looking at the Ms and SDs it is still the case that a lot of participants were likely at the highest point of 7 at baseline. Because the T0 FSC covariates also predicted the T1 FSC mediators, the mediators essentially act as a measurement of the within-subject change in the FSC constructs. As a result, there could have been reduced power to observe an effect of valence due to the inability to observe potential increases in valence for the percentage of participants that were at ceiling at baseline. I am not necessarily predicting an effect of valence would have been found otherwise, but it would be helpful to at least reference the potential ceiling effect on the valence item in the Discussion.

Yes, people always rate their future selves highly and this ceiling effect occurs in most uses of future self-valence. We added a bit in the limitations at line 452.

13. In lines 337-338, it is mentioned that the PROCESS macro ran four logistic regressions. I think the wording here might confuse some readers. To my understanding based on the earlier description of the mediation model and the PROCESS image included on OSF, one mediation model was performed with all 4 mediators included. That does mean four separate set of coefficients will be provided from the condition IV to the mediators, but some readers might interpret the wording here to imply that the mediation model was performed 4 times (with one mediator included at a time). Second, why would these predictions of the mediators be described as logistic regressions since the FSC mediators presumably are being treated as continuous variables (which is confirmed in the PROCESS image on OSF since the mediators are scale variables)?

We deleted the 4 regression equations because it is technical and confusing and then decided to rework the whole analytical strategy. This is needlessly confusing and we decided to take that part out. The mediation analysis is nice because it gave us all the results from our analysis and we integrated the technical function of mediation with the 3 hypotheses we tested.

To the second point, that is correct, the mediation models are logistic to the outcome,. We got rid of this wording at line ## as it is confusing and not necessary.

14. In Lines 448-449 the results of a pilot study are referenced for the first time. It was a little confusing as a reader because when I read that line it forced me to go back to the Methods/Results to see if that pilot study was mentioned before. I am not implying you need to reference those results before, but you may want to introduce the pilot study in these lines better so that it doesn’t confuse a reader.

We see that, usually pilots are mentioned. This observation occurred over the development of the exercise from feedback from our test subjects. We changed this in line 467.

15. In the Supplemental 2 Results, sometimes different condition names are used compared to the main document (e.g., Visual-VR). I recommend keeping the names consistent so it is clear to a reader comparing both results.

The supplemental results were checked for results and the condition names changed to their current form.

Reviewer #2

This study compared different ways of imagining a future self being successful with a specific focus on imagination and virtual reality integration. The authors looked at whether the impact of exposure to different future imaginings, and resultant ratings of similarity, connectedness and vividness to that future self on the transfer of intentions into behaviour.

The set up for the study was interesting and the authors offer sound possible interpretations of their findings. Where the study could be improved, from my perspective, is in the presentation of research questions and methodology; I found it difficult to follow this study and efforts could be put into place to more clearly explain what is expected and how the study was conducted. I provide more specific details below.

Introduction:

I found the introduction to be well written, logical and clear.

When describing the present study, I am curious why described the future self as having completed a long-terms personal goal (versus not having completed them). That is, why did the authors choose to take an approach versus an avoidance perspective. When imagining future selves, people may pursue a future self they would like to become (e.g., get my doctorate) but they may also want to engage in a behaviour (e.g., exercise) to avoid a future self they don’t want to realise (e.g., develop a chronic condition, get out of shape etc.). The chose to focus on a successful future self could be unpacked and justified. 

That is a great point that we are sensitive to but couldn’t include in our introduction. Our study wasn’t meant to parse approach from avoidance and approaching a goal was easier to integrate into the questions. To your point, we agree that in our study both approach or avoidance could have led to the formation of their new behavior and additionally approach or avoidance could have motivated participants to begin their new behavior. What was fascinating in the results is that some people were more motivated by less similarity to their future selves and some by feeling more connected to their future selves. We thought that labeling these results as characteristics of approach or avoidance was too much of a stretch over two measures that are already definitionally hazy. However, we do risk overinterpretation in the discussion by suggesting this may be a self-gap and self-continuity and at least point to new review of the self-gap/continuity discussion by Oyserman [44]beginning at line 377.

Method

It seems strange and exclusive that the VR software only allowed for the creation of male avatars. It begs the question, why are there VR software out there that are made to represent only one gender? The choice of this VR software limits the generalizability of the study, which is a big limitation.

We address that in the methods at line 157. This is a practical limitation of who the overall project. The software was designed for an intervention with juvenile delinquents, who are predominantly male. We put the male participants in the beginning and end of our discussion.

What inspired or served as the basis for the measurement of future self-continuity domains?

The basis is the Hershfield (2011) paper that suggests these measurements. We added further clarification

The authors explain why they controlled for imaginary ability by saying that “VR may aid participants with low imaginative ability”. This doesn’t see like a reason to control for imaginative ability. Wouldn’t you control for imaginative ability because it may influence how well one engages with imaginative tasks?

This was used as a control and as an exploratory moderation analysis added later. At line 172.

The authors should provide an explanation of what the steps of the perspective taking exercise are meant to accomplish. For example, for step 1, why did they have participants describe their future self having achieved a goal versus not having achieved it? For step 2, what do the authors think will come of the two chair exercise? And in step three, the behavioural intention? The reader can try to infer why these steps were chosen and what they are meant to accomplish but it would be clearer if the authors were explicit about why they chose these activities.

We restructured the methods to make what were trying to do clearer. This section starts at line 178. Relevant at overall at 178 and to perspective taking at 186. 

In the Imagine-VR condition, I understand that the participants did not use avatars, but then what did the virtual experience look like? Who did they see? What aspects about it were VR and why was this necessary?

That is true, the bird’s eye view in Fig1 may make it seem like the VR was not in the 1st person and its not obvious how people move from the present self to the future self in VR. We changed the methods, but had to make practical considerations to how much visual explanation we can provide. The second on the different conditions is now integrated in 3 paragraphs starting at line 185.

Overall, what was the intended function of the two-chair exercise? This is implied but could be made more

---

## [Editor Report · Decision Letter 1]

6 Jun 2024

Feeling connected but dissimilar to one’s future self reduces the intention-behavior gap

PONE-D-23-19069R1

Dear Dr. Ganschow,

We’re pleased to inform you that your manuscript has been judged scientifically suitable for publication and will be formally accepted for publication once it meets all outstanding technical requirements.

Kind regards,

André Rabelo, Ph. D.

Academic Editor

PLOS ONE
---

## [Editor Report · Acceptance letter]

18 Jun 2024

PONE-D-23-19069R1 

PLOS ONE

Dear Dr. Ganschow, 

I'm pleased to inform you that your manuscript has been deemed suitable for publication in PLOS ONE. Congratulations! Your manuscript is now being handed over to our production team.

Kind regards, 

on behalf of

Dr. André Luiz Alves Rabelo 

Academic Editor

PLOS ONE